# Polystyrene nanoplastics disrupt glucose metabolism and cortisol levels with a possible link to behavioural changes in larval zebrafish

Nadja R. Brun [1,2]*, Patrick van Hage [1], Ellard R. Hunting[3], Anna-Pavlina G. Haramis [4], Suzanne C. Vink [1], Martina G. Vijver [1], Marcel J.M. Schaaf [4] & Christian Tudorache [4]

Plastic nanoparticles originating from weathering plastic waste are emerging contaminants in aquatic environments, with unknown modes of action in aquatic organisms. Recent studies suggest that internalised nanoplastics may disrupt processes related to energy metabolism. Such disruption can be crucial for organisms during development and may ultimately lead to changes in behaviour. Here, we investigated the link between polystyrene nanoplastic (PSNP)-induced signalling events and behavioural changes. Larval zebrafish exhibited PSNP accumulation in the pancreas, which coincided with a decreased glucose level. By using hyperglycemic and glucocorticoid receptor (Gr) mutant larvae, we demonstrate that the PSNP-induced disruption in glucose homoeostasis coincided with increased cortisol secretion and hyperactivity in challenge phases. Our work sheds new light on a potential mechanism underlying nanoplastics toxicity in fish, suggesting that the adverse effect of PSNPs are at least in part mediated by Gr activation in response to disrupted glucose homeostasis, ultimately leading to aberrant locomotor activity.

[1] Institute of Environmental Sciences (CML), Leiden University, Leiden, The Netherlands. [2] Biology Department, Woods Hole Oceanographic Institution, Woods Hole, MA, USA. [3] School of Biological Sciences, University of Bristol, Bristol, UK. [4] Institute of Biology (IBL), Leiden University, Leiden, The Netherlands. *email: nbrun@whoi.edu

The global increase in plastic production and disposal has resulted in vast amounts of plastic debris in aquatic environments that pose both a burden and responsibility for the coming generations[1,2]. Assessing risk of plastic debris to the environment becomes progressively more complicated since plastic debris is broken down to micro- and ultimately nano-size scales through physical or digestive fragmentation[3,4]. Like the microplastics, the majority of the nano-sized particles accumulate in the gastrointestinal tract[5] or on the outer epithelium[6]. However, nanoplastics have the potential to cross epithelial barriers of vertebrates and have been reported to accumulate in the heart and brain of fish[7–9]. There remain considerable knowledge gaps in the mode of action of nanoplastics and the potential consequences at higher functional and organisational biological levels. Such knowledge is essential to ultimately allow for monitoring and predicting the consequences of the anticipated buildup of nanoplastics for the environment.

At the molecular level, nanoplastics can initiate stress response pathways such as oxidative stress, dysregulation of lipid and energy metabolism, and inflammation[6,10–13]. An inflammatory response of the innate immune system after exposure to polystyrene nanoparticles (PSNPs) is indicated by increased transcription of a key mediator in the neuromasts of zebrafish (*Danio rerio*) larvae[6], and increased necrosis, infiltration, and vacuolation in hepatocytes of adult zebrafish[13] and dark chub (*Zacco temminckii*)[14]. Additionally, at the cellular level, fathead minnow show activation of neutrophil function in the plasma when exposed to polystyrene and polycarbonate nanoplastics[11]. To date, however, at a higher level of biological organisation (e.g. organism or population level), it remains speculative if fish in nanoplastic-contaminated environments have a reduced host defence during a disease challenge. Also, nanoplastics can interact with lipid membranes[15] and disrupt metabolic processes in fish[10,12–14]. For example, dietary exposure to nanoplastics can lead to changes in metabolic profiles of liver and muscles of adult crucian carp (*Carassius carassius*)[10] and liver of adult zebrafish[13], while elevated cholesterol levels are found in the plasma of dark chub[14] and crucian carp[12], and the liver of zebrafish[13], indicating shifts in energy utilisation.

Recent studies have only started to unravel potential behavioural changes, a sensitive indicator of effects at the organism, population, or community level. Nanoplastic exposure in adult fish is associated with longer feeding time, lower activity, a stronger preference for staying close to conspecifics (shoaling behaviour), and reduced exploration of space[9,10]. Similarly, PSNPs exposure throughout zebrafish development leads to hypoactivity in larvae[7,16]. The mechanistic underpinning of these PSNP-induced behavioural changes in fish remains to a large extent elusive, but can potentially be tied to neurological or metabolic effects. For example, the shoaling behaviour is thought to be mediated by neurotransmitters, specifically the dopaminergic system[17]. Changes in metabolic rate are widely accepted as a proxy for stress response, and are correlated with behavioural endpoints such as exploration or swimming activity[18,19]. Furthermore, coping with stress results in different patterns in both serotonergic activity[20] and cortisol[21] as part of a complex set of feedback interactions between the hypothalamus, the pituitary gland, and interrenal tissues (HPI-axis).

Cortisol is the main endogenous glucocorticoid in teleosts and most mammals, and seems to play a key role in a wide variety of processes including innate immune responses, intermediary metabolism, and behaviour[22–26]. Elevated cortisol secretion is a major hallmark of stress response[19]. Under stressful conditions, cortisol mainly acts through the intracellular glucocorticoid receptor (zebrafish protein: Gr, zebrafish gene: *gr*), whereas under basal condition, its effects are mainly mediated by the mineralocorticoid receptor[27,28]. Increased cortisol levels have been observed to coincide with alterations in behaviour, particularly locomotion[29–31]. In this context, zebrafish larvae are increasingly used as a model organism to study the molecular aspects of behavioural changes in fish, in part because zebrafish harbour only one *gr* gene (in contrast to most other fish species that contain two). In zebrafish larvae, cortisol levels have been observed to increase in response to a physical stressor (swirling) as early as 4 days post fertilisation (dpf)[32]. Moreover, elevated cortisol levels induced by stress, starvation, or glucocorticoids[33] can stimulate gluconeogenesis and thereby increase blood glucose levels[19]. A hallmark gene of gluconeogenesis is *phosphoenolpyruvate carboxykinase 1* (*pck1*), which encodes the rate-limiting enzyme in this process. Cortisol and gluconeogenesis may be reciprocally regulated, as hyperglycaemic zebrafish embryos exhibit increased cortisol levels[34]. Hence, for fish and many other vertebrates, this suggests a complex interplay between cortisol, gluconeogenesis, and behaviour that is likely prone to environmental contaminants such as PSNPs.

By considering an unappreciated set of responses at molecular signalling and behavioural levels, we suggest here the involvement of disrupted energy and cortisol metabolism in inducing an adverse behavioural effect in fish larvae after exposure to PSNPs. Taking advantage of the zebrafish as an emerging model organism in metabolic disease and behavioural research, we have used a *gr* mutant and a *pck1* transgenic bioluminescence reporter

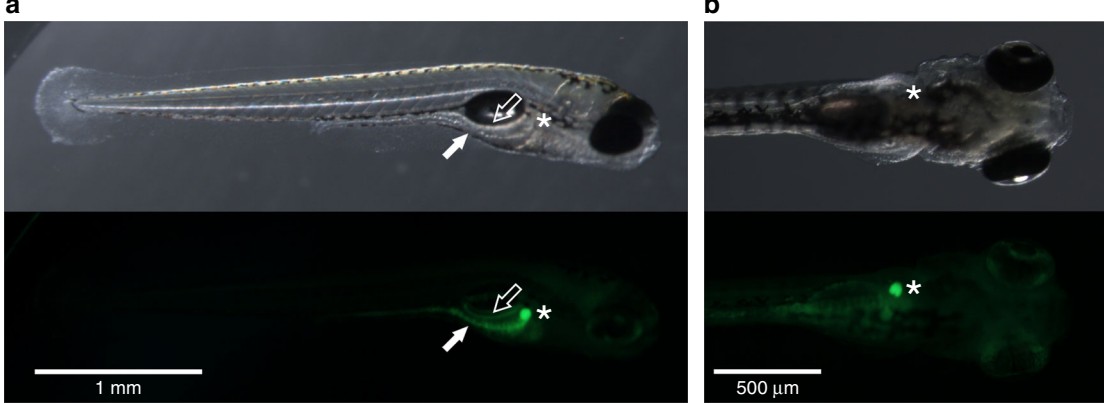

**Fig. 1** Biodistribution after PSNP exposure in zebrafish larvae from 72 to 120 hpf. **a** Representative pictures of PSNP accumulation in the intestine (lateral view, solid arrow), exocrine pancreas (lateral view, empty arrow), and **b** gallbladder (ventral and lateral view, asterisk) of wild-type zebrafish larvae. The experiment was conducted on three separate occasions with ten biologically independent replicates each

zebrafish line to disentangle the consecutive events elicited by PSNPs. We present evidence that PSNPs induce changes in both glucose and cortisol levels, as well as in gluconeogenesis activity in zebrafish larvae. Given the direct involvement of cortisol in increased activity[30], we have subsequently examined behavioural changes by measuring distance moved during alternating light–dark cycles as a common behavioural trigger in fish using wild-type, hyperglycemic, and *gr* mutant larvae exposed to PSNPs. We confirmed that glucose homoeostasis, as well as the Gr, are likely mediating the observed changes in behaviour.

## Results

**Biodistribution and physiological response**. To determine target organs, zebrafish larvae were imaged after exposure to fluorescently labelled PSNPs. PSNPs accumulated in the intestine, exocrine pancreas, and gallbladder of exposed larvae (Fig. 1). The highest PSNP concentration tested (20 mg L$^{-1}$) did not significantly affect the growth of zebrafish larvae at 120 hours post fertilisation (hpf) ($F(2, 25) = 1.65$, $p = 0.2151$), although a slight reduction in the mean length of PSNP-exposed larvae was observed (Supplementary Fig. 1a). By contrast, swim bladder development was significantly affected ($F(14,87) = 33.65$, $p < 0.0001$), with 50.1% of treated wild-type larvae having inflated swim bladders, compared to 91.4% of the controls (Supplementary Fig. 1b). Interestingly, the larvae with an inflated swim bladder did not show any reduction in swim bladder size following exposure to PSNP (Supplementary Fig. 1c), suggesting that once inflation was initiated, the process was not further affected.

**Effects on cortisol levels**. To investigate the involvement of cortisol in the response to PSNP exposure, cortisol levels in whole larvae were measured. Cortisol was significantly increased in the wild-type strain AB/TL after exposure to PSNP ($F(3, 20) = 14.86$, $p < 0.0001$). The mean cortisol level for 2 mg L$^{-1}$ PSNPs ($M = 98.39$, SD = 16.58) and 20 mg L$^{-1}$ PSNPs ($M = 100.2$, SD = 10.06), but not for 0.2 mg L$^{-1}$ PSNPs, was significantly higher in comparison to the control ($M = 57.39$, SD = 9.46; Fig. 2a). Similar results were found for the wild-type strain ($gr+/+$) used to create $gr-/-$, $t(8) = 3.58$, $p = 0.0072$. When co-exposed to

glucose (Fig. 2b) and in the $gr-/-$ larvae (Fig. 2c), exposure to PSNP did not alter cortisol levels, indicating the involvement of both glucose and activation of Gr in response to PSNP exposure.

**Effects on glucose metabolism**. Several endpoints indicative of activation of metabolic processes to support energy-demanding activities were assessed in control and PSNP-exposed larvae. A two-way analysis of variance showed that the effect of PSNP on glucose levels was significant, $F(3, 63) = 21.89$, $p < 0.0001$, as well as the zebrafish strain factor, $F(3, 63) = 180.5$, $p < 0.0001$ (Fig. 3a). Post-hoc analysis using Bonferroni adjusted alpha levels of 0.05 indicated that PSNPs significantly reduced the whole-body glucose level in 5 dpf AB/TL larvae at the highest dose ($M = 0.09598$, SD $= 0.01169$) in comparison to the control ($M = 0.1493$, SD $= 0.006285$), adj. $p < 0.0001$ (Fig. 3a). Similar results were found for $gr+/+$, $F(3, 63) = 21.89$, $p < 0.0001$. After Bonferroni correction, the highest dose group did have significantly lower glucose levels ($M = 0.04424$, SD $= 0.01227$) than the control ($M = 0.08926$, SD $= 0.01588$), adj. $p = 0.0003$. Larvae missing the $gr$ ($gr-/-$) or wild-type larvae with pharmacologically reduced $gr$ activity (mifepristone) did not appear to have affected glucose levels after PSNP exposure. Insulin staining, marking the pancreatic islet, showed that PSNP exposure leads to a significant reduction of the size of the insulin expression domain (Fig. 3b, Supplementary Fig. 2; $t(12) = 2.65$, $p = 0.0212$). These findings suggest that bioaccumulated PSNPs may affect glucose metabolism. The activity of the promoter driving the expression of a gene encoding a key enzyme for gluconeogenesis, *pck1*, was significantly increased with increasing PSNP concentration ($F(8, 32) = 53.85$, $p < 0.0001$), likely to counteract the reduced glucose level (Fig. 3c). Post-hoc comparison indicated that the mean score of *pck1* activity for the highest PSNP dose ($M = 13,270$, SD $= 3020$) was significantly higher in comparison to the control ($M = 1428$, SD $= 670.7$). Blocking Gr using the receptor antagonist mifepristone partially inhibited the increase in promoter activity in PSNP-exposed larvae ($M = 7679$, SD $= 1611$, adj. $p < 0.0008$; Fig. 3c), suggesting that increased gluconeogenesis due to PSNP exposure is at least partially mediated through Gr activation. Despite the decreased activity in comparison to PSNP treatment only, co-exposure to mifepristone still led to a dose-dependent increase in *pck1* activity. Furthermore, upon PSNP exposure,

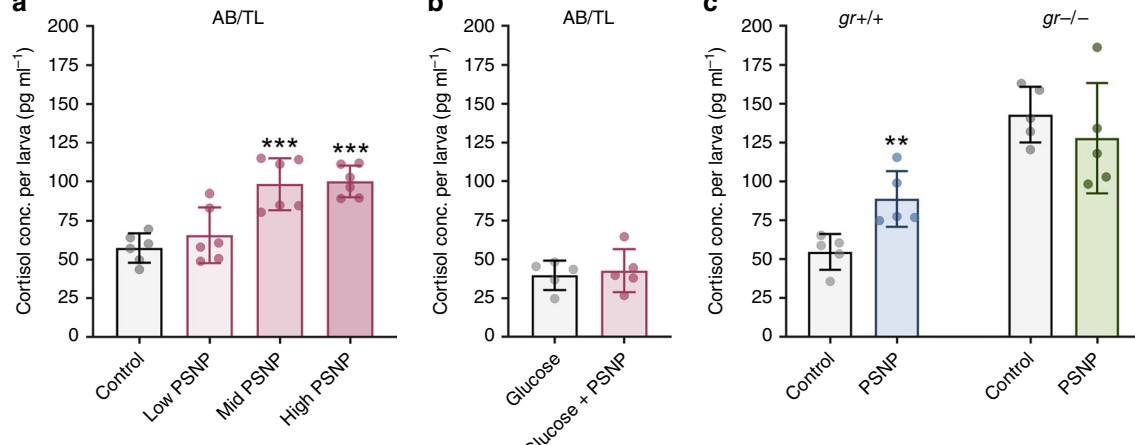

**Fig. 2** Cortisol levels in zebrafish larvae exposed to PSNPs. **a** Cortisol levels increase in a dose-dependent manner after exposure to 0.2 mg L$^{-1}$ (low), 2 mg L$^{-1}$ (mid), and 20 mg L$^{-1}$ (high) PSNP from 72 to 120 hpf ($n = 6$, pool of 15 zebrafish per sample). **b** No difference in cortisol levels is detected when reared in 40 mM glucose from 72 to 120 hpf and exposed to PSNPs (glucose + PSNP) from 72 to 120 hpf ($n = 5$, pool of 15 zebrafish per sample). **c** Glucocorticoid receptor mutants ($gr-/-$) have a high cortisol baseline level due to the negative feedback loop and no difference in PSNP-exposed larvae is detected. The $gr+/+$ represent the wild-type strain used to create the $gr-/-$ strain ($n = 5$, pool of 15 zebrafish per sample). Values are presented as mean ± SD. Asterisks indicate significant differences to controls (*$p < 0.05$, **$p < 0.01$, and ***$p < 0.001$)

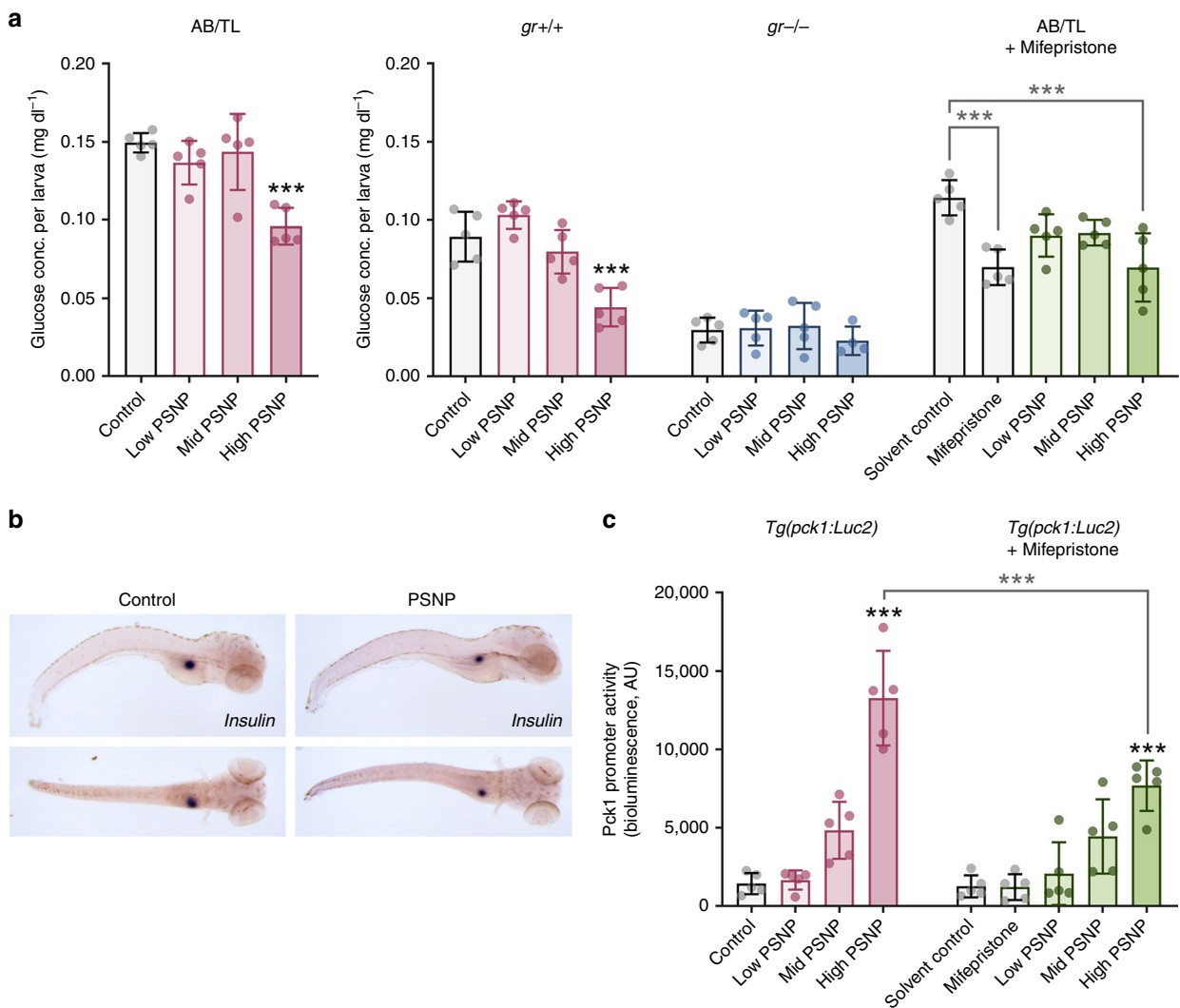

**Fig. 3** Effects of PSNPs on glucose metabolism in larval zebrafish. **a** Dose-dependent response showing glucose concentrations in zebrafish embryos wild-type (AB/TL and *gr+/+*), glucocorticoid receptor mutant (*gr−/−*), and AB/TL supplemented with the glucocorticoid receptor antagonist mifepristone (1 μM; *n* = 4–5, pool of ten zebrafish per sample). **b** Representative image of insulin staining of the pancreas in control and 20 mg L$^{-1}$ PSNP-exposed embryos (*n* = 7, biologically independent replicates). **c** Dose-response showing *pck1* promoter dynamics in *Tg(pck1:Luc2)* zebrafish larvae at 120 hpf after exposure to 0.2 mg L$^{-1}$ (low), 2 mg L$^{-1}$ (mid), and 20 mg L$^{-1}$ (high) PSNPs from 72 to 120 hpf (*n* = 5, pool of three zebrafish per sample). Values are presented as mean ± SD. Asterisks indicate significant differences to controls (*$p < 0.05$, **$p < 0.01$, and ***$p < 0.001$)

*glucose-6-phosphatase a* (*g6pca*) expression was significantly upregulated ($t(7) = 4.042$, adj. $p < 0.0294$) and *fibroblast growth factor 21* (*fgf21*) expression ($t(8) = 4.864$, adj. $p < 0.0072$) as well as *lactate dehydrogenase a* (*ldha*) expression ($t(8) = 7.581$, adj. $p < 0.0001$) were downregulated, and also the transcript of the *solute carrier family 6 member 4* (*slc6a4*) encoding for a membrane protein that transports the neurotransmitter serotonin from synaptic spaces into presynaptic neurons was significantly downregulated ($t(8) = 3.562$, adj. $p = 0.0444$; Supplementary Fig. 3). Both *pck1* and *g6pca* are rate-limiting enzymes in gluconeogenesis and glycogenolysis, respectively, while *ldha* catalyses the final step of anaerobic glycolysis and *fgf21* plays an important role in regulating hepatic lipid and glucose homoeostasis. In summary, exposure to the highest PSNP concentration significantly decreased glucose levels, despite the simultaneous increase in glycolytic and gluconeogenic activity, which is dependent on Gr activation.

The *gr* mutant zebrafish larvae had a lower glucose level ($M = 0.02953$, SD $= 0.007925$) than wild-type zebrafish under basal conditions ($M = 0.1493$, SD $= 0.006285$), with no additional

effect when exposed to PSNPs (Fig. 3a). Congruently, exposure to 1 μM mifepristone reduced the glucose concentration in larval zebrafish ($M = 0.06979$, SD $= 0.01139$) in comparison to the solvent control ($M = 0.1141$, SD $= 0.01129$) and no additional effect of exposure to PSNPs was observed (Fig. 3a). These results indicate that inactivation of Gr results in decreased glucose levels under basal conditions and that these levels are not further aggravated upon PSNP exposure.

**Behavioural response.** An increase in activity can be triggered by a variety of mechanisms, including modulation of neuronal activity and rapid elevation of plasma cortisol[19,20]. Moreover, stress-induced increases in cortisol levels can fuel modulation of neuronal activity[19,35–37]. Here, we tested the effect of PSNP exposure and different zebrafish strains on the distance moved during the dark challenge phase. PSNP exposure induced a significant alteration in locomotion during the dark challenge phase of the behavioural assessment ($F(1, 759) = 46.02$, $p < 0.0001$) and the effect of zebrafish strains yielded an F ratio of $F(4, 759) = 9.543$, $p < 0.0001$. PSNP exposed wild-type larvae

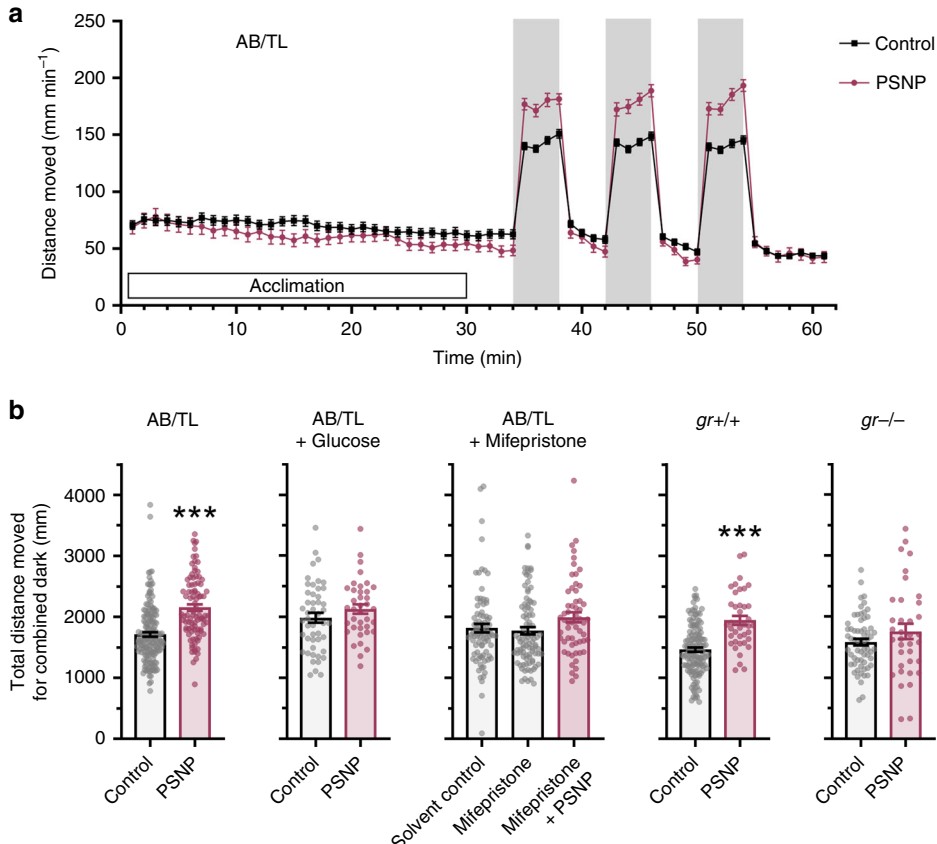

**Fig. 4** PSNP exposure effects on larval behaviour. **a** Locomotor activity of control ($n = 165$, biologically independent replicates) and PSNP ($n = 91$, biologically independent replicates) exposed AB/TL larvae throughout behavioural tracking with 34 min acclimation and three times 4-min dark challenge phase followed by 4-min light recovery phases. Activity was measured as distance moved (mm) in 1 min per individual larvae. **b** Cumulative activity (mm) in all dark phases (three times 4 min) tracked for individual larvae. PSNP-exposed zebrafish larvae (wild-type AB/TL and $gr+/+$) showed increased locomotor activity in the dark challenge (control AB/TL: $n = 165$, PSNP AB/TL: $n = 91$, control $gr+/+$: $n = 136$, PSNP $gr+/+$: $n = 40$). Addition of 40 mM glucose diminished the hyperactivity in PSNP-exposed wild-type zebrafish (control AB/TL + glucose: $n = 48$, PSNP AB/TL + glucose: $n = 38$) and similarly, $gr-/-$ zebrafish and mifepristone co-exposed zebrafish exhibit an activity pattern alike the control (control $gr-/-$: $n = 63$, PSNP $gr-/-$: $n = 36$, control AB/TL + mifepristone: $n = 84$, PSNP AB/TL + mifepristone: $n = 58$). Data points represent biologically independent replicates from at least three independent experiments and the error bars indicate the mean ± SEM. Asterisks indicate significant differences to controls (*$p < 0.05$, **$p < 0.01$, and ***$p < 0.001$)

exhibited a distinct hyperactivity ($M = 2129$, SD $= 560.1$) in comparison to the control ($M = 1701$, SD $= 486.3$), adj. $p < 0.0001$ (Fig. 4a and b). A similar response is shown for the wild-type strain $gr+/+$ (Fig. 4b) used to create the $gr-/-$, where the mean difference of the PSNP-exposed larvae was significantly higher ($M = 1944$, SD $= 462.4$) than in the $gr+/+$ control group ($M = 1463$, SD $= 411.4$), adj. $p < 0.0001$. This hyperactivity in the dark phase was suppressed in $gr-/-$ larvae with no significant difference between control and exposed group (Fig. 4b), and similar results were observed when co-exposing wild-type larvae to the Gr antagonist mifepristone (Fig. 4b), indicating that the observed changes in behavioural responses to the dark challenge induced by PSNP exposure were mediated by cortisol-activated Gr. In addition, when offering excess amounts of glucose in the medium, exposure to PSNPs does not evoke hyperactivity during the dark challenge, suggesting that the distorted energy metabolism is fuelling the behavioural change. The particle control group ($TiO_2$) showed no difference in activity in comparison to the control (Supplementary Fig. 4), implying that the effect observed here is rather a plastic compound than a nanoparticle effect. During the light recovery phases, no effect of PSNP exposure on the larval behaviour was observed (Supplementary Fig. 5).

## Discussion

Nanoplastics are an emerging, yet poorly understood environmental contaminant of global concern. Identifying molecular modes of action is therefore essential to characterise the toxic potential of nanoplastics and will, if linked to the organismal level of response, advance our abilities to assess the risk they pose to the environment. Here, we identify a set of interdependent events for a nanoplastics-induced stress response (Fig. 5) and show that a disruption in glucose homoeostasis and increase in cortisol secretion coincide with behavioural changes in zebrafish larvae.

The localisation of a contaminant in target tissues can support the identification of toxic mechanisms. At the time of exposure (72–120 hpf), PSNPs accumulate in neuromasts[6] and the jaw movement of the zebrafish larvae is already developed, thereby facilitating ingestion of particulate matter from the surrounding medium. As reported earlier, the gastrointestinal tract is thus an important organ of accumulation from where the particles can spread through the circulatory system from which they are cleared by receptor-specific endocytosis in fish[38] and can accumulate in various organs exhibiting a particularly slow depuration from the intestine and pancreas[5–8,13]. Similarly, in our study, PSNPs concentrated in the gastrointestinal tract as well as the gallbladder and the exocrine pancreas. In developing zebrafish at

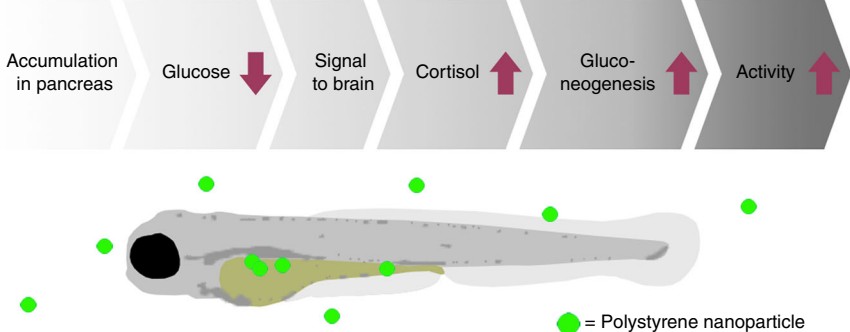

**Fig. 5** Simplified overview of the measured metabolic and behavioural responses along the zebrafish HPI-axis in response to stress induced by PSNPs. Arrows indicate an increase (↑) or decrease (↓) of the main interlinked stages of glucose metabolism assessed in this study that suggest a bottom-up driven chain of events where glucose homoeostasis fuels cortisol secretion and behaviour

5 dpf, the exocrine and endocrine pancreas are likely not as well separated in their function as in later stages[39]. It is thus conceivable that PSNPs aggregating in the exocrine pancreas could affect the endocrine pancreas, resulting in lower glucose level, which is signalled to the brain where the HPI-axis is activated leading to cortisol secretion. Cortisol then activates Grs, which are distributed heterogeneously in various tissues throughout zebrafish development[40].

We explored the potential effects of PSNP exposure on glucose homoeostasis and cortisol secretion and observed that both processes are affected in PSNP-exposed wild-type zebrafish larvae. Specifically, whole-body glucose levels, as well as insulin expression, were decreased. This likely resulted in an increase in cortisol production, which in turn activates *g6pca* and *pck1* gene expression, two genes involved in glycogenolysis and gluconeogenesis. Despite the increased gluconeogenic activity, the glucose stores were depleted. The state of low glucose level can elicit a stress response, thereby increasing cortisol secretion through the Gr[41]. The well-known direct effects of cortisol on gluconeogenesis[42] indicate that cortisol is fuelling the *pck1* activity. Here, we interpret that increased cortisol secretion in response to decreased glucose levels mediates the effect of PSNP (Fig. 5). This is supported by the compensatory effect of glucose on increased cortisol levels (Fig. 2b). At a later stage of development of larval zebrafish (6 dpf), stress-induced elevated cortisol levels are linked with reduced feeding, further aggravating low glucose levels, and generating a negative feedback mechanism[43]. Ultimately, exposure to excessive cortisol during early life stages can be translated to effects in adulthood including permanent epigenetic modification of the glucocorticoid receptor and direct elevated basal cortisol levels, defective tailfin regeneration, and immunoregulation[44,45].

Increased levels of cortisol during a stress event are known to result in the reallocation of energy away from investment activities such as growth and reproduction towards short-term activities such as locomotion and tissue repair[19]. Although a reduced growth rate is commonly observed during toxicant exposure and periods of elevated cortisol[19], the PSNP-exposed larvae were of similar size as the control larvae (Supplementary Fig. 1a). The exposure period of 2 days used in this study might be too short to capture this reallocation of energy resources at the level of growth, yet it is conceivable that PSNP exposure could result in impaired growth rates if longer exposure times are considered.

We observed PSNP-induced alterations of glucose and cortisol levels to coincide with an altered behavioural response of the zebrafish larvae, visible as hyperactivity upon sudden darkness. Chronic PSNP exposure throughout development has also been observed to cause hypoactivity, potentially due to distortion of neural development and function[7,16]. In the present study, larval

fish were exposed between 3 and 5 dpf and our results indicate that PSNP-induced changes in cortisol levels have the potential to modulate how fish respond to a challenge (darkness), since wild-type larvae exposed to PSNP exhibited hyperactivity in the dark challenge and PSNP exposure did not affect the behaviour of *gr* mutant larvae. Increased hyperactivity in the dark phase at 4 dpf has been observed previously after cortisol exposure between 1 and 48 hpf[30], and in another study, hyperactivity was observed in the light phase at 4 dpf upon injection of cortisol at the 1-cell stage[46]. In addition to differences in methodological approaches between different studies (e.g. exposure, duration), these studies collectively seem to suggest that PSNPs- or cortisol-induced behavioural responses can differ depending on context and stage of development.

We consider two mechanisms that are most likely underlying these cortisol-driven behavioural alterations. First, cortisol can interfere with the electrical activity of brain cells to alter the level of important molecules, including neurotransmitters, enzymes, or receptors. Glucocorticoid effects on the brain are highly complex and brain region-, dose- as well as time-dependent[35–37]. Recently such interferences with the neural system have been observed for fish like trout, medaka, and zebrafish[47–49], and could potentially lead to aberrant stress-coping mechanisms (e.g. stress recovery patterns and anxiety-related behaviours). In support of this, we observed that neurotransmitter activity might indeed be affected as the gene transcribing the membrane protein that transports the neurotransmitter serotonin (*slc6a4*) is downregulated (Supplementary Fig. 3). Second, altered glucose levels can fuel cortisol secretion with inherent changes to energy metabolism and availability to sustain activity. In addition to the contribution of Gr activation to the observed heightened locomotor activity during the dark challenge, we explored the contribution of a dysregulated metabolic rate. The addition of 40 mM glucose as known reducer of *pck1* expression in larval zebrafish[50] coincided with diminished cortisol levels (Fig. 2b) and hyperactivity in larvae exposed to PSNP (Fig. 4b), providing strong support for distorted energy metabolism as mechanism fuelling the behavioural responses in this study. While future research should uncover whether the elevated cortisol increased locomotor activity observed in this study results from interference with the neural system, increased mobilisation of glucose to sustain the movement[18], or a combination thereof, our results thus point towards a bottom-up driven chain of events where decreased glucose levels fuel cortisol secretion and an aberrant behavioural response (conceptually depicted in Fig. 5).

Uncertainties that remain call for a better understanding of events that are uncharted in this study. Here, the plastic component of the nanosized particle was the most likely source of behavioural alterations as we did not observe behavioural effects

in our control-particle experiment (TiO$_2$, Supplementary Fig. 4). Similarly, one of the most widely applied plasticising compound (bisphenol A) has been associated with altered zebrafish larval locomotion[51]. However, in natural systems, it is conceivable that behavioural effects are ultimately dependent on developmental stage and the result of a combination of a particle and plastic effect. For instance, foreign nanoparticles (e.g. plastic, gold) can disrupt the epithelial layers, accumulate in the circulatory system[8,52], and induce inflammatory responses[6] in larval zebrafish, potentially hinting a disruption of the HPI-axis. Likewise, nanoparticles can increase plasma cortisol levels in adult fish[53–55]. However, in larval fish, neither cortisol levels nor locomotion is altered by metal-based nanoparticlces[52,56]. We also observed reduced inflation of the swim bladder in PSNP-exposed wild-type and $gr-/-$ larvae (Supplementary Fig. 1b), suggesting mechanisms operating independently from Gr activity are likely present. Several other molecular events relevant to the proposed chain were also not considered here. For instance, molecular stress-induced feedback mechanism between cortisol and glucose levels and potential neural interferences remain uncertain, albeit our gene expression data indicate that serotonin is affected (Supplementary Fig. 3). Another key question is how these molecular events partition in dominating behavioural effects during progressive stages of development. These knowledge gaps could guide future efforts resolving the hitherto unnoticed effects of plastics on glucose metabolism and cortisol-induced changes in fish behaviour.

In conclusion, we used developing zebrafish to illustrate that one of the most abundant plastic polymers, polystyrene, can in its nanoparticle form disrupt glucose homoeostasis with concurrent activation of the stress response system. Our data provide evidence that increased cortisol secretion is likely tied to increased locomotion during challenge phases. This study thereby began to unravel a currently overlooked set of interlinked events in fish triggered by exposure to PSNPs and hence encourages future characterisation of uncharted molecular mechanisms underpinning the effects of PSNPs. The presented mode of action triggered by PSNPs and inherent adverse outcome could potentially serve studies aimed at predicting the effects of nano-sized plastic particles on aquatic communities. While stress-induced activation of HPI-axis is evident in many organisms confronted with various forms of stress, organisms at the early stage of developing are likely to be more sensitive to alterations in energy metabolism. The current buildup of plastic will likely contribute to the existing plethora of pollutants that interfere with the stress-axis and behaviour of fish, potentially affecting their performance and interactions with their immediate environment.

## Methods

**Materials.** Green fluorescent PSNPs (25 nm; 1.05 g cm$^{-3}$) internally dyed with Firefli$^{TM}$ Fluorescent Green (468/508 nm) were obtained from Thermo Fisher Scientific (Waltham, U.S.). Its characteristics in egg water were previously assessed in our group using transmission electron microscopy (TEM) and dynamic light scattering (DLS), showing a stable hydrodynamic diameter of 19 nm over the first 24 h[6]. Titanium dioxide nanoparticles (TiO$_2$; 15–24 nm, 3.9 g cm$^{-3}$) were obtained from the JRC Nanomaterials Repository (JRC ID: JRCNM01005a) and the size confirmed using TEM (Supplementary Fig. 6). D-(+)-Glucose and RU486 (mifepristone) were purchased from Sigma Aldrich (St. Louis, U.S.). A stock solution of 5 mM mifepristone was prepared in ethanol (200 proof, molecular biology grade, ≥99.45%, Sigma Aldrich). The molecular biology kits used are indicated below.

**Animal husbandry and larvae exposure.** Zebrafish were handled in compliance with the directives of the local animal welfare committee of Leiden University (License number: 10612) and maintained according to standard protocols (http://ZFIN.org). All protocols adhered to the international guidelines specified by the EU Animal Protection Directive 2010/63/EU. As only early life stage zebrafish were used in experiments, no specific additional project authorisation was needed. Zebrafish wild-type (AB/TL strain) eggs were obtained from natural spawning

family crossings, $Tg(pck1:Luc2,cryaa:mCherry)^{s952}$ (ref. [57]) (hereafter named $Tg(pck1:Luc2)$) from single crossings with TL strain, and $gr+/+$ or $gr-/-$ (or $gr^{s357}$)[24], respectively, from single self-crossings. Fertilised eggs were selected within the 2- to 8-cell stage of the cleavage period and incubated in aerated egg water (60 µg mL$^{-1}$, Instant Ocean Sea Salt; Sera GmbH, Heinsberg, Germany) at 28.5 ± 0.5 °C with a 10:14-h dark:light cycle until sampling at 120 hpf. The daily upkeep included rinsing every 24 h with aerated egg water, and the removal of coagulated embryos up to the start of the exposure at 72 h.

The exposure concentration of 20 mg L$^{-1}$ PSNPs was derived from an initial dose-response analysis representing a no-effect concentration for mortality (Supplementary Fig. 7). Mortality was assessed by following the protocol of the Fish Embryo Acute Toxicity Test (FET) and adapted to an exposure window from 72 to 120 hpf. Hatched larvae were exposed in 24-well plates (1 larva per well filled with 2 mL of solution). Ten exposure concentrations between 10 and 100 mg L$^{-1}$ and a control solution consisting of egg water were tested (four replicates, ten larvae per replicate). For some of the assays, concentrations of 2 and 0.2 mg L$^{-1}$ were tested additionally. The particle control experiment was performed using TiO$_2$ nanoparticles (19.5 nm) at a concentration of 38.603 mg L$^{-1}$ to match the particle number of PSNPs (25 nm) at a concentration of 20 mg L$^{-1}$. All exposures were started after hatching, at 72 hpf, as the chorion can represent a physical barrier for nanoparticles[5,58], and lasted until 120 hpf with one medium exchange at 96 hpf. Co-exposure to 1 µM mifepristone or 40 mM glucose and respective controls (mifepristone, glucose, or solvent control only) were performed where appropriate.

**Physiological response and PSNP biodistribution.** After 2 days of exposure to 20 mg L$^{-1}$ PSNP, at 120 hpf, ten larvae per dose group were anaesthetised in 0.02% tricaine (MS222, Sigma Aldrich) and imaged from a lateral and ventral perspective using a Leica MZ16FA fluorescence stereomicroscope equipped with a digital camera (DFC420). The lateral bright-field images were scored for swim bladder development (presence or absence) as well as swim bladder surface area and larval length, which were measured using ImageJ[59]. Biodistribution of PSNPs was detected using the green fluorescence laser filter while imaging both lateral and ventral sides. All measurements were repeated three times with larvae from separate breeding events ($n = 10$ per group and breeding event).

**Larval cortisol measurement.** Cortisol concentration of control larvae and three exposure concentrations (0.2, 2, 20 mg L$^{-1}$) were measured after 2 days of exposure at 120 hpf according to a protocol previously described by Tudorache et al.[21]. Briefly, 15 larvae per replicate were pooled and 6 replicates per group sampled. The larvae were snap-frozen in liquid nitrogen and then homogenised in 500 µL of phosphate-buffered saline (PBS) using a bullet blender. Cortisol was extracted with two volumes of diethyl ether three times. After evaporation of diethyl ether overnight, the cortisol was redissolved in 0.2% bovine serum albumin (BSA) in PBS. Larval cortisol measurements were carried out using a cortisol ELISA kit (Abnova KA1885) according to the manufacturer's instructions and absorbance read at 450 nm using a plate reader (Tecan Infinite M1000 PRO).

**In vivo luciferase reporter assay.** To assess the effect of PSNPs on gluconeogenesis, $Tg(pck1:Luc2)$ larvae were exposed to 0, 0.2, 2, and 20 mg L$^{-1}$ PSNPs in 48-well plates with 1 mL exposure volume per well. Co-exposure to 40 mM glucose and 1 µM mifepristone was performed as well (1 larva per well). In the $Tg(pck1:Luc2)$[57] fish, the cytosolic phosphoenolpyruvate carboxykinase (pck1) promoter is labelled with firefly luciferase gene luc2. Phosphoenolpyruvate is used as the starting substrate for gluconeogenesis and transcriptional alterations of pck1 are predominantly expressed in the liver and kidneys. At 120 hpf, larvae were washed twice in egg water and three larvae were pooled to one replicate with five replicates sampled per group. Larvae were lysed in a final volume of 100 µL of egg water using a point sonicator (Qsonica) with five pulses at an amplitude of 20% and stored on ice until centrifugation at 13,000 r.p.m. for 3 min at 4 °C. Subsequently, 90 µL of the supernatant was transferred to a 96-well plate (Thermo Fisher, flat bottom, white) and 50 µL of Steady-Glo (Promega) was added. The plate was incubated in the dark for 1 h after which the bioluminescence was quantified using a plate reader (Tecan Infinite M1000 PRO).

**Whole-mount in situ hybridisation.** Wild-type zebrafish larvae used to assess insulin expression by whole-mount in situ hybridisation were raised in 0.003% 1-phenyl-2-thiourea (PTU; Sigma-Aldrich) added no later than at 24 hpf to prevent pigmentation. Control and PSNP-exposed larvae were anaesthetised on ice and fixed in 4% paraformaldehyde at 120 hpf ($n = 7$ per treatment group). Subsequently, whole-mount in situ hybridisation was performed according to the standard protocol (2008)[60]. The riboprobe against ins has been previously described [61] and was a kind gift from Dr. Rubén Marín-Juez. Larvae were imaged using a Leica MZ16FA fluorescence stereomicroscope.

**Larval glucose measurement.** Glucose levels in control and exposed wild-type (AB/TL and $gr+/+$) and $gr-/-$ larvae were measured by a glucose colourimetric assay kit (Cayman Chemical). After 48 h exposure (at 120 hpf), ten larvae per replicate ($n = 5$) were pooled in 55 µL of glucose buffer and homogenised using

a Bullet blender (Next Advance Inc.). The homogenate was centrifuged at 13,000 r.p.m. for 10 min at 4 °C and 50 μL of the supernatant was transferred to a transparent 96-well plate. Glucose standards were prepared according to the manufacturer's protocol. The enzymatic reaction was initiated by adding 50 μL of the enzyme mixture to both samples and the standard. After an incubation time of 20 min at 37 °C in the dark, absorbance was measured at 514 nm by a plate reader (Tecan Infinite M1000 PRO). Fluorescence values were corrected by subtracting measurements from control reactions without sample and glucose levels were interpolated from standard curves.

**Quantitative polymerase chain reaction**. Gene expression changes of transcripts related to glucose metabolism (*fgf21*, *g6pca1*, *slc2a2*, *ldha*), serotonin transport (*slc6a4*), and oxidative stress (*cat*) were analysed as described previously[62] and primer sequences are listed in Supplementary Table 1. Briefly, 15 wild-type larvae per replicate (*n* = 5) were snap-frozen at 120 hpf after 48 h of exposure to egg water or 20 mg L$^{-1}$ PSNPs, respectively. RNA was isolated using an RNeasy Mini kit (Qiagen) and RNA quantified on a NanoDrop ND-1000 Spectrophotometer (Nanodrop Technologies Inc., U.S.) while quality was visually verified on an agarose gel. RNA was reverse transcribed using the Omniscript$^{TM}$ Reverse Transcriptase kit (Qiagen, The Netherlands), Oligo-dT primers (Qiagen), and RNase inhibitor (Promega). The samples were denatured for 5 min at 95 °C and then amplified using 40 cycles of 15 s at 95 °C and 45 s at 60 °C followed by quantitation using a melting curve analysis post-run. Amplification and quantification were done with the CFX96 Biorad system. Fold induction was calculated by normalising $C_T$ values of the target gene to the $C_T$ value of the housekeeping gene *β-actin* (=$\Delta C_T$) and then normalised to the untreated control ($\Delta C_T$ untreated – $\Delta C_T$ treated).

**Larval behaviour**. For the behavioural assessment, AB/TL, *gr+/+*, and *gr−/−* larvae of 48 hpf in age were distributed to a polystyrene 48-well plate (Corning Costar, Corning), one in each well. At 72 hpf, the controls (*n* = 24) received 1 mL of fresh egg water while two independent treatment groups (*n* = 24) received either 20 mg L$^{-1}$ PSNPs or 38.603 mg L$^{-1}$ TiO$_2$ nanoparticles in egg water. In the case of co-treatment with mifepristone, as solvent control was included with a replicate size of 16 per treatment group and experiment. Treatment groups were randomly distributed and all larvae kept in the well plate until 120 hpf, with a medium replacement after 24 h. At 120 hpf, larvae without swim bladder were removed and then the individual distance moved in an alternating light/dark test was quantified as an indication of stress using the DanioVision$^{TM}$ observation chamber (Noldus Inc.)[63]. Observations started after 3 h after onset of light when larval activity is at a stable level for several hours[64]. During observation period larvae were exposed to the following stressor chain: after an acclimation period of 30 min in the illuminated chamber, a light baseline of 4 min was tracked before the stressor chain started with a dark challenge (4 min) and a light recovery (4 min). The stressor chain was repeated three times in total. The experiment was repeated three times with cohorts from separate breeding events. Video data were recorded with 30 frames per second via a high-speed infrared camera. Obtained data were analysed with the supplied software EthoVision XT® 12 (Noldus. Inc.).

**Statistics and reproducibility**. All statistical analyses were conducted using Graph Pad Prism 8.0. The data of all assays (cortisol, luciferase assay, glucose assay, quantitative polymerase chain reaction (qPCR), and larval locomotor activity) were tested for deviation from the Gaussian ideal using the Shapiro–Wilk normality test. For assays comprising a full factorial design with an equal number of treatments and different zebrafish strains (glucose, *pck1* activity, locomotion), a two-way ANOVA was carried out. The difference between the solvent control and mifepristone-treated larvae was tested separately using a one-way ANOVA. For the cortisol assay, comprising different numbers of treatments per zebrafish strain, a one-way ANOVA or *t*-test, respectively, was performed per zebrafish strain. All ANOVA comparisons were followed by multiple comparisons using the Bonferroni correction to adjust the critical values. For all other assays with one tested concentration for the wild-type zebrafish strain (qPCR, insulin expressing area), treatment groups were compared by a two-tailed *t*-test (with a Satterthwaite approximation for unequal sample sizes) and in case of a multiple gene factor (qPCR) *p*-values adjusted by multiplication with the total number of genes tested. A *p*-value < 0.05 (*t*-test) or a Bonferroni-corrected *p*-value < 0.05 (ANOVA) was considered statistically significant. Replicates consisted of a pool of larvae (3–15, depending on the assay) except for morphological, in situ hybridisation, and locomotion analysis where one replicate represents one larva. The data are presented as mean ± SD with single data points (replicates) superimposed on the graph.

**Reporting summary**. Further information on research design is available in the Nature Research Reporting Summary linked to this article.

## Data availability

All source data underlying the graphs presented in the main figures are reported in Supplementary Data 1. All data and materials produced by this study are available from the corresponding author upon request.

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

## Acknowledgements

We thank Natalia Novik and Laurie Mans for technical assistance during glucose assay and in situ hybridisation, respectively, Rubén Marín-Juez for providing the *ins* riboprobe, and John J. Stegeman for his helpful comments on the manuscript. The research described in this work was supported by the Dutch research council NWO (MGV; 864.13.010).

## Author contributions

N.R.B. conceived the experiments and coordinated the study. N.R.B., M.J.M.S., A.-P.G.H., and C.T. participated in the design of the study. N.R.B., P.H., and S.C.V. performed the experiments. N.R.B. and P.H. analysed the data. N.R.B. wrote the manuscript, and M.J.M.S., E.R.H., and C.T. contributed significantly to earlier drafts of the manuscript. M.G.V. provided logistical support and resources. All authors contributed to scientific discussions and editing of the content of previous versions of the manuscript and approved submission of the final draft.

## Competing interests

The authors declare no competing interests.
