## [Peer Review File · Communications Biology]

Reviewers' comments:

Reviewer #1 (Remarks to the Author):

The manuscript is well written, well organized, and covers interesting content that will be of interest to researchers from numerous disciplines when published. The manuscript describes a study worthy of publication in communications biology. However, the authors must first refine the manuscript to better reflect the scope and inference of the study. Depending on the author's capacity to clarify/rectify the outlying concerns of the reviewer, this could entail minor or major revisions. The current body of literature pertaining to zebrafish larval behavior and neurochemical mechanisms that manifest phenotypes is rife with confusion and over generalizations. The authors should see this as an opportunity to help reduce this chaos by being as specific as possible when describing zebrafish larval behavioral phenotypes, while staying well confined in the inference space allowed to them by their experimentation.

No effort was placed to discuss how exactly elevated cortisol modulates behavior in zebrafish larvae even though this is a principal conclusion of the study. This affects the potential impact of this study, leaving it primarily as a toxicological study of the effects of microplastics. The authors could improve the potential impact of this paper by providing a targeted discussion on the potential mechanisms of cortisol related behavioral modulation.

The authors provide convincing evidence that cortisol plays an important role in modulating zf locomotor activity. However, the perspective they provide in the manuscript must be more open to alternative mechanisms that have been previously identified, specifically in terms of microplastic exposure. For example, others have also found that acetylcholine esterase activity is modulated by microplastics exposure (Chen et al. 2017. *Sci. Tot. Env.* 15:584-585), which has clearer and better documented links to hypo/hyper activity than does cortisol.

Numerous other behavioral studies with zf have found microplastic exposure induced hypoactivity in larvae, a finding that contradicts the findings of this study. There is no effort in the discussion to reconcile this difference. Behavior in early lifestage zebrafish is not well understood and is complicated by the fact that the brain and nervous system are not yet fully developed. Behavioral phenotypes and their mechanistic underpinnings are likely to change as the brain develops, and this happens quickly in larval zebrafish. Given that the central thesis of the study is on behavior, a more detailed discussion regarding larval zebrafish behaviors seems necessary, and this can potentially be used as support to explain why hyperactivity ensued microplastic exposure in this the present study while hypoactivity has been observed in multiple other studies.

It is not immediately clear why larvae that are in an advanced state of fasting (Line 256; i.e., energetically compromised) would have higher locomotor activity.

Lines 58 – 60: "This suggests a mechanistic link to observed behavioral changes such as longer feeding time, lower activity, stronger shoaling behavior, and reduced exploration of space^{9,10}". It is not clear what the mechanistic link is, particularly in regards to the specific behavioral phenotypes that are listed. The authors should be as specific as possible.

Lines 64 – 66: "The dysregulation of the innate immune response, the intermediary metabolism, and natural behavior point to a response mediated by cortisol due to the well-known key role of this hormone in these processes¹⁷". This reference does not support that cortisol has a well-known key role in natural behavior in larval zebrafish. It is not wise to make generalizations regarding zebrafish behavior, as there are numerous phenotypes that are regulated by dissimilar pathways, and there is

clear evidence that mechanisms of adult behavior are dissimilar to mechanisms of larval behaviors. The authors should make use of literature that is more relevant to the age group and if possible, specific light:dark locomotor activity phenotypes used in the present study.

Lines 67 – 70: Include appropriate references for GR and MR function.

Line 73: replace “. . .because of zebrafish. . .” with “. . . because zebrafish. . .”

Line 76: “increased cortisol levels can be directly related to increased activity^{20–22}”. Again, the authors need to take more care in explaining cortisol mediated hyperactivity in reference to the phenotype being observed. For example, Ref 20 indicates increased locomotor activity in the light and no difference in the dark (contradicts present findings), Ref 21 found increased locomotor activity in the dark, but not in the light (supports present findings), and ref 22 found hyperactivity in the light (contradicts present findings). There is much contradiction in the literature which the authors have chosen not to discuss, weakening the inference of their findings to informed readers, and perhaps giving a falsehood of determinate evidence to a more naïve reader.

Lines 92 – 94: “Given the direct involvement of cortisol in increased activity²⁰, we have subsequently determined behavioral changes by measuring distance moved during alternating light-dark cycles. . .”. Ref 20 is perhaps not the most appropriate reference for this statement, as they did not demonstrate a direct involvement of cortisol in increased activity.

Line 142: “Increased activity is a hallmark in fish with elevated cortisol levels”. Provide reference. Increased activity is also a hallmark of fish exposed to variety of neuroactive compounds as well, which has nothing to do with cortisol or energy metabolism. A naïve reader may believe all cases of elevated activity are a function of elevated cortisol, which is not true. Sentence should be reworded.

Line 146: “indicating that this behavioral effect is mediated by Gr”. The authors need to be more careful when referring to their behavioral phenotype and the role of Gr. This can easily be solved by increasing the specificity of their statements throughout the manuscript. For example, a more accurate statement would be: “indicating that the observed changes in behavior induced by PSNP exposure were mediated by Gr”.

Line 146: “. . . similar results were observed when co-exposing wild-type larvae to the Gr antagonist mifepristone (Figure S4, SI), . . .”. These results are not shown in Figure S4. They also should be illustrated in Fig. 4 along with the rest of the results for behavior.

Lines 214 – 217: “After 5 dpf, zebrafish larvae begin to transition from yolk-feeding to fasting whereby when unfed, glucose levels progressively decrease, while pck1 activity increases until 7 dpf to combat the decreasing glucose in a growing system²³. The PSNP-exposed larvae may thus be in an advanced state of starvation. . .”. This doesn’t make sense because the authors used 5 dpf fish for their behavioral trials. The authors should reconsider this statement.

Figure 4. The authors should make it clear what their replication represents. It is not clear whether n refers to individual larvae, 1 min bins for each fish (i.e., repeated measurements), or 1 min bins for each fish for each repeated dark stage.

Lines 377 – 379: It is not clear why a t-test was used in this scenario, when an omnibus tests (i.e., anova) is more appropriate. Their experimental design includes interactive effects that were not tested appropriately (i.e., by use of t-tests), greatly weakening the inference of their results. Their experimental design is not fully factorial, which is problematic. For example, what would be the effect

of glucose and mifepristone treatment on $gr^{+/+}$ and $gr^{-/-}$ mutants? Moreover, it is not specified if a p-value correction was used to account for the multiple t-tests carried out. The way the description in the methods reads is that the authors essentially carried out a post-hoc without first testing for the main effects of PSNP and mutant.

Reviewer Comments and Suggestions for Improvement.

Reviewer #1 (Remarks to the Author):

The manuscript is well written, well organized, and covers interesting content that will be of interest to researchers from numerous disciplines when published. The manuscript describes a study worthy of publication in communications biology. However, the authors must first refine the manuscript to better reflect the scope and inference of the study. Depending on the author's capacity to clarify/rectify the outlying concerns of the reviewer, this could entail minor or major revisions. The current body of literature pertaining to zebrafish larval behavior and neurochemical mechanisms that manifest phenotypes is rife with confusion and over generalizations. The authors should see this as an opportunity to help reduce this chaos by being as specific as possible when describing zebrafish larval behavioral phenotypes, while staying well confined in the inference space allowed to them by their experimentation.

This comment is very much appreciated and we took the opportunity to clarify the current knowledge and knowledge gaps in the mechanistic understanding of behavioural traits in zebrafish larvae.

We also elaborate now on different potential mechanisms underlying varying behavioural traits in the 3rd paragraph of the introduction.

No effort was placed to discuss how exactly elevated cortisol modulates behavior in zebrafish larvae even though this is a principal conclusion of the study. This affects the potential impact of this study, leaving it primarily as a toxicological study of the effects of microplastics. The authors could improve the potential impact of this paper by providing a targeted discussion on the potential mechanisms of cortisol related behavioral modulation.

We appreciate this comment and have enriched the discussion with the following sentences:

Increased hyperactivity in the light phase at 4 dpf has been observed previously after cortisol exposure between 1 and 48 hpf³¹, and in another study, hyperactivity was observed in the light phase at 4 dpf upon injection of cortisol at the 1-cell stage⁴⁵. Glucocorticoids modulate behaviour through different mechanisms. It is likely that elevated cortisol increased locomotor activity by interfering with the neural system leading to aberrant stress-coping styles (stress recovery patterns and anxiety-related behaviours)¹⁹ or by mobilizing glucose and thereby energy to sustain the movement.

The authors provide convincing evidence that cortisol plays an important role in modulating zf locomotor activity. However, the perspective they provide in the manuscript must be more open to alternative mechanisms that have been previously identified, specifically in terms of microplastic exposure. For example, others have also found that acetylcholine esterase activity is modulated by microplastics exposure (Chen et al. 2017. Sci. Tot. Env. 15:584-585), which has clearer and better documented links to hypo/hyper activity than does cortisol.

Alternative mechanisms which could be responsible for hypo/hyperactivity are now acknowledged in the discussion (L279ff).

“There is evidence that prolonged PSNP-induced hypoactivity in the dark may be related to the inhibition of acetylcholinesterase¹⁶, an enzyme responsible for inactivation of the neurotransmitter acetylcholine.”

Numerous other behavioral studies with zf have found microplastic exposure induced hypoactivity in larvae, a finding that contradicts the findings of this study. There is no effort in the discussion to

reconcile this difference. Behavior in early lifestage zebrafish is not well understood and is complicated by the fact that the brain and nervous system are not yet fully developed. Behavioral phenotypes and their mechanistic underpinnings are likely to change as the brain develops, and this happens quickly in larval zebrafish. Given that the central thesis of the study is on behavior, a more detailed discussion regarding larval zebrafish behaviors seems necessary, and this can potentially be used as support to explain why hyperactivity ensued microplastic exposure in this the present study while hypoactivity has been observed in multiple other studies.

We agree the MS gains strength when including an attempt to reconcile apparent disparities referenced in the MS. To this end, we articulated a major difference to previous studies. In the current study, an exposure window from 3 to 5 days post fertilization was used whereas previous studies exposed throughout development. Exposure throughout development might affect development by clogging the pores of the chorion and thereby reducing nutrient and oxygen exchange. Moreover, the development of the brain and nervous system might be affected. Thus, In the current study, we disentangled early developmental effects from larval effects by exposing freshly hatched larval zebrafish that have all major organs developed.

Specifically, the discussion is extended with (L277ff):

“Chronic PSNP exposure throughout development has also been observed to cause hypoactivity, potentially distorting neural development and function^{7,16}. There is evidence that prolonged PSNP-induced hypoactivity in the dark may be related to the inhibition of acetylcholinesterase¹⁶, an enzyme responsible for inactivation of the neurotransmitter acetylcholine. In the present study, The larval fish were exposed between 3 and 5 dpf and our results indicate that PSNP-induced s and inherent changes in cortisol levels have the potential to modulate how fish respond to a challenge (darkness), since wild-type larvae exposed to PSNP exhibited hyperactivity in the dark challenge and PSNP exposure did not affect the behaviour of gr mutant larvae.”

It is not immediately clear why larvae that are in an advanced state of fasting (Line 256; i.e., energetically compromised) would have higher locomotor activity.

We agree that this might be counter-intuitive. Fasting/Starvation can induce increased cortisol secretion (as mentioned in L80). Given the speculative nature of this hypothesis, we have therefore removed it from the conclusions.

Lines 58 – 60: “This suggests a mechanistic link to observed behavioral changes such as longer feeding time, lower activity, stronger shoaling behavior, and reduced exploration of space^{9,10}”. It is not clear what the mechanistic link is, particularly in regards to the specific behavioral phenotypes that are listed. The authors should be as specific as possible.

This comment is very much appreciated and we elaborate now on the mechanistic link to the specific behavioural phenotypes potentially involved in L56ff.

“Recent studies have only started to unravel potential behavioural changes, a sensitive indicator of effects at the organism, population, or community level. Nanoplastic exposure in adult fish is associated with longer feeding time, lower activity, a stronger preference for staying close to conspecifics (shoaling behaviour), and reduced exploration of space^{9,10}. Similarly, PSNP exposure throughout zebrafish development leads to hypoactivity in larvae^{7,16}. The mechanistic underpinning of these PSNP-induced behavioural changes in fish remains to a large extent elusive but can potentially be tied to neurological or metabolic effects. For example, the shoaling behaviour is thought to be mediated by neurotransmitters, specifically the dopaminergic system¹⁷. While

differences in metabolic rate is a widely accepted proxy for stress response and are correlated with behavioural endpoints such as exploration or swimming activity.^{18,19} “

Lines 64 – 66: “The dysregulation of the innate immune response, the intermediary metabolism, and natural behavior point to a response mediated by cortisol due to the well-known key role of this hormone in these processes¹⁷”. This reference does not support that cortisol has a well-known key role in natural behavior in larval zebrafish. It is not wise to make generalizations regarding zebrafish behavior, as there are numerous phenotypes that are regulated by dissimilar pathways, and there is clear evidence that mechanisms of adult behavior are dissimilar to mechanisms of larval behaviors. The authors should make use of literature that is more relevant to the age group and if possible, specific light:dark locomotor activity phenotypes used in the present study.

Thank you for the careful reading. We now use more careful wording to avoid generalizations. The existing reference (indicating that 5-day old embryos treated with 1 μ M cortisol exhibit overrepresented genes involved in immune system processes) is bolstered with more of the specific references for larval zebrafish including Griffiths et al. (2012) and Van Den Bos et al. (2017) demonstrating the correlation between cortisol level and locomotion in the light/dark assay in larval zebrafish.

Existing reference:

Gross, K. L. & Cidlowski, J. A. Tissue-specific glucocorticoid action: a family affair. Trends Endocrinol. Metab. 19, 331–9 (2008).

New references added:

Facchinello, N. et al. Nr3c1 null mutant zebrafish are viable and reveal DNA-binding-independent activities of the glucocorticoid receptor. Sci. Rep. 7, 1–13 (2017).

Griffiths, B. B. et al. A zebrafish model of glucocorticoid resistance shows serotonergic modulation of the stress response. Front. Behav. Neurosci. 6, 1–10 (2012).

Van Den Bos, R. et al. Further characterisation of differences between TL and AB Zebrafish (Danio rerio): Gene expression, physiology and behaviour at day 5 of the larval stage. PLoS One 12, 1–15 (2017).

Chatzopoulou, A. et al. Transcriptional and metabolic effects of glucocorticoid receptor α and β signaling in zebrafish. Endocrinology 156, 1757–1769 (2015).

Lines 67 – 70: Include appropriate references for GR and MR function.

The following referenced have been added:

Faught, E. & Vijayan, M. M. The mineralocorticoid receptor is essential for stress axis regulation in zebrafish larvae. Sci. Rep. 1–11 (2018).

de Kloet, E. R., Meijer, O. C., de Nicola, A. F., de Rijk, R. H. & Joëls, M. Importance of the brain corticosteroid receptor balance in metaplasticity, cognitive performance and neuro-inflammation. Front. Neuroendocrinol. 49, 124–145 (2018).

Line 73: replace “. . .because of zebrafish. . .” with “. . . because zebrafish. . .”

Replaced

Line 76: “increased cortisol levels can be directly related to increased activity^{20–22}”. Again, the authors need to take more care in explaining cortisol mediated hyperactivity in reference to the phenotype being

observed. For example, Ref 20 indicates increased locomotor activity in the light and no difference in the dark (contradicts present findings), Ref 21 found increased locomotor activity in the dark, but not in the light (supports present findings), and ref 22 found hyperactivity in the light (contradicts present findings). There is much contradiction in the literature which the authors have chosen not to discuss, weakening the inference of their findings to informed readers, and perhaps giving a falsehood of determinate evidence to a more naïve reader.

We appreciate this input and now highlight the discrepant findings on effects of cortisol in the light/dark assay (L74ff).

“Increased cortisol levels have been observed to coincide with different alterations in behaviour, particularly locomotion^{30-32.}”

Furthermore, we come back to this in the discussion (L276ff).

“We observed PSNP-induced alterations of cortisol levels to result in an altered behavioural response of the zebrafish larvae, visible as hyperactivity upon sudden darkness. Chronic PSNP exposure throughout development has also been observed to cause hypoactivity, potentially due to distortion of neural development and function^{7,16}. There is evidence that prolonged PSNP-induced hypoactivity in the dark may be related to the inhibition of acetylcholinesterase¹⁶, an enzyme responsible for inactivation of the neurotransmitter acetylcholine. In the present study, larval fish were exposed between 3 and 5 dpf and our results indicate that PSNP-induced changes in cortisol levels have the potential to modulate how fish respond to a challenge (darkness), since wild-type larvae exposed to PSNP exhibited hyperactivity in the dark challenge and PSNP exposure did not affect the behaviour of gr mutant larvae. Increased hyperactivity in the light phase at 4 dpf has been observed previously after cortisol exposure between 1 and 48 hpf³¹, and in another study, hyperactivity was observed in the light phase at 4 dpf upon injection of cortisol at the 1-cell stage⁴⁵. Glucocorticoids modulate behaviour through different mechanisms. It is likely that elevated cortisol increased locomotor activity by interfering with the neural system leading to aberrant stress-coping styles (stress recovery patterns and anxiety-related behaviours)¹⁹ or by mobilizing glucose and thereby energy to sustain the movement.”

Lines 92 – 94: “Given the direct involvement of cortisol in increased activity²⁰, we have subsequently determined behavioral changes by measuring distance moved during alternating light-dark cycles. . .”. Ref 20 is perhaps not the most appropriate reference for this statement, as they did not demonstrate a direct involvement of cortisol in increased activity.

We agree and have replaced the reference with the following reference:

Steenbergen, P. J., Bardine, N. & Sharif, F. Kinetics of glucocorticoid exposure in developing zebrafish: A tracer study. Chemosphere 183, 147–155 (2017).

Line 142: “Increased activity is a hallmark in fish with elevated cortisol levels”. Provide reference. Increased activity is also a hallmark of fish exposed to variety of neuroactive compounds as well, which has nothing to do with cortisol or energy metabolism. A naïve reader may believe all cases of elevated activity are a function of elevated cortisol, which is not true. Sentence should be reworded.

Thank you for the input, we have reworded the sentence and added references accordingly.

“An increase in activity can be triggered by a variety of mechanisms including modulation of neuronal activity and rapid elevation of plasma cortisol^{18,21}. Moreover, stress-induced increases in cortisol levels can fuel modulation of neuronal activity and behavioural responses^{21,36}.”

Line 146: “indicating that this behavioral effect is mediated by Gr”. The authors need to be more careful when referring to their behavioral phenotype and the role of Gr. This can easily be solved by increasing the specificity of their statements throughout the manuscript. For example, a more accurate statement would be: “indicating that the observed changes in behaviour induced by PSNP exposure were mediated by Gr”.

The careful reading is appreciated and the specificity of our wording has been increased throughout the manuscript.

Line 146: “. . . similar results were observed when co-exposing wild-type larvae to the Gr antagonist mifepristone (Figure S4, SI), . . .”. These results are not shown in Figure S4. They also should be illustrated in Fig. 4 along with the rest of the results for behavior.

We apologize for having referred to the wrong figure in the Supplementary. The correct Figure would have been Figure S5. In the revised version of the manuscript, this figure is now part of Figure 4.

Lines 214 – 217: “After 5 dpf, zebrafish larvae begin to transition from yolk-feeding to fasting whereby when unfed, glucose levels progressively decrease, while pck1 activity increases until 7 dpf to combat the decreasing glucose in a growing system²³. The PSNP-exposed larvae may thus be in an advanced state of starvation. . .”. This doesn’t make sense because the authors used 5 dpf fish for their behavioral trials. The authors should reconsider this statement.

This comment is appreciated and we have reconsidered the statement leading to its deletion.

Figure 4. The authors should make it clear what their replication represents. It is not clear whether n refers to individual larvae, 1 min bins for each fish (i.e., repeated measurements), or 1 min bins for each fish for each repeated dark stage.

Figure 4b is displaying the cumulative distance swum in all three dark challenge phases (three times 4 minutes). The wording has been adjusted for clarification and replicate numbers added in all figure descriptions.

“Cumulative activity (mm) in all dark phases (three times 4 minutes) tracked for individual larvae.”

Lines 377 – 379: It is not clear why a t-test was used in this scenario, when an omnibus tests (i.e., anova) is more appropriate. Their experimental design includes interactive effects that were not tested appropriately (i.e., by use of t-tests), greatly weakening the inference of their results. Their experimental design is not fully factorial, which is problematic. For example, what would be the effect of glucose and mifepristone treatment on gr+/+ and gr-/- mutants? Moreover, it is not specified if a p-value correction was used to account for the multiple t-tests carried out. The way the description in the methods reads is that the authors essentially carried out a post-hoc without first testing for the main effects of PSNP and mutant.

We agree that a two-way ANOVA is more appropriate to compare the total distance moved and glucose levels among the different zebrafish strains. The statistical tests have been re-run and the methods (L464ff) and results section (L112ff) revised. We considered performing a REML on the cortisol levels which contains data of multiple strains and variable number of concentrations per strains but refrained from it after conferring with mathematicians. A one-way ANOVA or t-test, respectively, was performed per strain.

Also, we followed the APA style giving degrees of freedom, t-values and exact p-values for all the significant tests in the results section.

Reviewer #2

General:

The manuscript provided for review was entitled “Polystyrene nanoplastics disrupt glucose metabolism resulting in cortisol-induced behavioral changes in larval fish”. This was a well written manuscript that utilized an innovative suite of techniques to evaluate the linkages between nanoparticle exposure and bioaccumulation with glucose metabolism and cortisol induced behavioral changes in zebrafish larvae. The methods used were appropriate for the study with some minor corrections could be easily followed by other scientists. A couple of weaknesses were identified that unfortunately affect the strength of the manuscript and the stated findings. The first is the low samples numbers used in most experiments (particularly the first dose response curves conducted to define doses throughout the rest of the study), the second is more significant to the scope of the paper. To my understanding, no particle control was used yet the authors make many strong statements on the impacts of polystyrene nanoplastics and the links to glucose, cortisol, and cortisol induced behaviors. The broader impacts of the study as stated in the introduction is also clearly about plastics. However, without a non-plastic particle control, the authors can’t distinguish between nanoplastics and any other nanoparticle (of which there are potentially many in aquatic environments). Some suggestions are given in the methods on how the authors could handle this and certainly some justification and discussion on limitations is needed. This does not invalidate the results obtained, but it does change the interpretation and impact from my perspective. If a non-plastic nanoparticle control had been included this would be a compelling study. I don’t doubt the “effect” observed in the present study, certainly nano-plastics must be doing something since it is a foreign particle that can get inside an organism and cause irritation/damage (and this would cause an immune response). It is just not clear if any particle would do the same. In the end this is a small point because we have a lot of microplastics and thus nanoplastics entering the aquatic environment. I would just like to see clarification/caution so that the data are not interpreted as “only nanoplastics have these effects”, since that was not established in this study.

We greatly appreciate the reviewer’s constructive and supportive comments. All the comments were helpful and contributed to improving the manuscript.

Most importantly, we have now additionally performed a particle control (Titanium dioxide nanoparticle of 20 nm in diameter) experiment showing that the locomotor activity of larval zebrafish was not modulated. We also substantiate the discussion with references to previously performed locomotor assays using nanoparticle exposures and discuss the limitations.

Lastly, although larger sample numbers would certainly increase the power, we detected significant differences in all assays, which in itself points towards sufficiently large enough sample sizes.

Specific Text Edits and comments by section:

Title: This overstates the evidence so I would recommend a minor revision that highlights the data in the manuscript but not overstating the links. Suggestion below.

Possible link between polystyrene nanoplastic exposure on glucose metabolism and cortisol-induced behavioral changes in larval zebrafish

I’m actually not sure you can say more than nanoparticle since there was no particle control.

Thank you for identifying that the title did not correctly reflect the content of the study. Taking into account the reviewer's suggestion, the additional experiment using TiO₂ nanoparticles, and aiming generally for more careful wording, we have adjusted the title as follows:

Polystyrene nanoplastics disrupt glucose metabolism and cortisol levels with a possible link to behavioural changes in larval zebrafish

Abstract:

General comment: The abstract needs some editorial work.

line 16:aquatic environments with an ~~yet~~ unknown mode of actions in aquatic organisms.

The editing efforts of the reviewer are much appreciated, acknowledged, and corrected accordingly.

line 16-17: Recent studies ~~suggest hint~~ that internalized nanoplastics ~~may can~~ disrupt processes related to energy metabolism.

Acknowledged and corrected accordingly.

line 17-19: Such disruptions can be ~~particularly~~ crucial for organisms during ~~the~~ early stages of development and may ultimately lead to changes in behavior.

Acknowledged and corrected accordingly.

line 19-20: Here, ~~we investigated the~~causally link ~~between~~ polystyrene nanoplastic (PSNP)-induced signaling events ~~that lead and~~ behavioral changes.

Acknowledged and corrected accordingly.

line 20-21: "potentially" suggests that you could not determine if the decreased glucose levels were due to PSNP accumulation in the pancreas. If this is the case you can't say that you found a causal relationship in the previous sentence. You may have 2 lines of evidence in your study but the authors should be cautious in drawing cause and effect for processes that can be affected by many things that may or may not have been controlled in the study, such as the particle type. Consider revising your statements to highlight your data rather than the "possible" or "potential" links since this may be speculative. In line reference discussed: Larval zebrafish exhibited a decreased glucose level, potentially resulting from PSNP accumulation in the pancreas.

Thank you for the careful reading. We agree that that the causal link to the accumulation in the pancreas is not proven in this study. The word 'potentially' and 'causally' was removed and sentences rephrased.

line 24: Please revise to ensure that the data is represented but not overstated. The data and relationships highlighted are interesting so worthy of publication, but not fully elucidated as yet. Minor changes can keep this in accordance with the data presented in the manuscript.

...we demonstrate that the PSNP23 induced disruption in glucose homeostasis ~~consecutively results correlated with~~ ~~in~~ increased cortisol secretion ~~and leading to~~ hyperactivity in challenge phases.

Acknowledged and corrected accordingly.

line 24-25: Again, be careful with overstating your evidence. Your findings and interpretations are interesting and provide some evidence to support your postulation, but additional work should be done before you “prove” these linkages given the processes investigated. A minor edit will improve this. See below:

Therefore, the adverse effect of PSNPs ~~appears to be~~ ~~is~~ mediated by Gr activation in response to disrupted glucose homeostasis.

Acknowledged and corrected accordingly.

line 25-27: Please revise or remove the concluding statement in the abstract. The first half of the sentence repeats the last. In the second half, it is unclear how this study would be used as a blueprint for assessment of emerging environmental threats. Do you mean that glucose homeostasis and cortisol secretion should be evaluated in those assessments? That would be a challenging as I am sure the authors realize, since many factors can influence global stress hormones and glucose regulation and metabolism.

~~This study thus frames a previously unknown train of events elicited by PSNPs in fish and provides a blueprint for future assessment of emerging environmental threats.~~

Acknowledged. The last sentence has been revised as follows (L24ff):

“Our work sheds novel light on a potential mechanism underlying nanoplastics toxicity in fish, suggesting that the adverse effect of PSNPs are mediated by Gr activation in response to disrupted glucose homeostasis, ultimately leading to aberrant locomotor activity.”

Introduction:

Line 30-32: The ~~tremendous-global~~ increase in plastic production and disposal ~~has~~ resulted in ~~an~~ ~~undeniable presence of~~ vast amounts of plastic debris in aquatic environments that poses both a burden and responsibility for the coming generation.

The use of “tremendous” and “undeniable” should be avoided. The authors should try not to overstate or unnecessarily amplify the problem. It is a big problem and it will be for a long time. Just keep the language clear and concise wherever possible.

Acknowledged and corrected accordingly.

Line 36-39: This seems to be two separate points and a very long sentence. You already mentioned that nanoplastics are very small so you can also simplify by removing this piece. You have added considerable nominalizations by combining these thoughts, so split into two to improve the clarity. This enhances your points that are otherwise lost.

However, ~~due to their small size,~~ nanoplastics ~~also~~ have the potential to cross epithelial barriers of vertebrates, and ~~have been reported to~~ accumulate in ~~organs such as the~~ heart and brain in fish. ~~yet~~

There remain considerable knowledge gaps in the mode of action of nanoplastics and its the potential consequences at higher functional and organizational biological levels.

Acknowledged and corrected accordingly.

Line 61-62: Behavioral changes in a single species are actually not a good predictor/indicator of ecosystem level disturbance or impact. You could suggest organism and population level, possibly community depending on the species relevance and functional role in the community, but not ecosystem. Please delete.

Acknowledged and 'ecosystem' has been deleted.

Line 64-66: As an entry to a new paragraph, this sentence doesn't seem to settle on a point.

Unfortunately, this makes no sense. Please revise to clarify the point here. It seems like a list of things you plan to discuss, but not well linked.

In line reference: "The dysregulation of the innate immune response, the intermediary metabolism, and natural behavior point to a response mediated by cortisol due to the well-known key role of this hormone in these processes."

This input is appreciated and the entry sentence to the new paragraph has been revised as follows:

"Cortisol is the main endogenous glucocorticoid in teleost and most mammals and seems to a key role in a wide variety of processes, including innate immune responses, intermediary metabolism, and natural behavior²²⁻²⁶."

Line 66-67: You did not make a clear point in the prior sentence so you should remove "furthermore" and just introduce cortisol and why it is an element of this study.

~~Furthermore,~~ Elevated cortisol secretion is a major hallmark of the response to stress.

Acknowledged and corrected accordingly.

Results:

Line 111: "exposure, cortisol levels in whole larvae ~~where were~~ measured. Cortisol was ~~which were~~ significantly increased...."

Acknowledged and corrected accordingly.

Line 121: "These findings suggest that bioaccumulated PNSPs ~~may~~ affect glucose metabolism."

You need to give yourself some wiggle room here since you did not control for non-plastic particles, so this may not be a "plastics" response. This is a compromised edit. In the absence of controls, I would prefer to see "bioaccumulated nanoparticle" without the inclusion of plastics in any statement of cause and effect or even relationships.

Thank you for the comment. We appreciate and go with the more careful wording. We believe that the additional control experiment justifies using PSNP instead of 'bioaccumulated nanoparticle'.

Figure 2: The Y-axis of 2A should be corrected to be the same increments as B and C. This is minor but since the Y-axis is otherwise the same it would be better to also make the increments the same for easy reference. The fluorescent green dots in panel C are really hard to see so please consider a different contrasting color. Please clarify sample number per group in the figure legend.

We apologize for having overseen the dissimilar increments and have adjusted the figure. Additionally, the size of the data points was increased, darker border colour of the dots added, and the sample number indicated per group in the figure legend.

Figure 3: If allowed by editorial staff, split this into 3 figures (panel A, panel B, and panel C). The fluorescent green dots are again a challenge. Please consider an alternative. Panel D is not needed, this is well explained in the text and very little is gained by a bar graph for this data. This should be moved to supplementary materials. If figure number is restricted removal of at least D to SI will allow a little more room for the other panels. In the figure legend please clarify sample number per panel and group.

We agree with this input and have moved panel D to the supplementary as well as increased the dot sizes and darker border colour.

Figure 4: revise title. Figure 4. PSNP exposure **effects on** larval behavior. Figure legend: remove the range for sample number since this is ambiguous. Just make this clear across experiments and groups.

The title is revised and sample number indicated per exposure group.

Discussion:

The discussion is well written and well supported by the available scientific literature. Caution should be taken in some of the bolder statements, based on earlier comments and issues raised in the methods section.

We appreciate this comment and have made adjustments throughout the discussion.

Figure 5: This is a bit crude and unnecessary. Schematics like this can be useful for a graphical abstract but this does not add much to the paper. If this figure is maintained the title has to be changes to remove “causally”.

We appreciate this comment. However, we would like to keep the scheme in the manuscript. It serves well to explain the suggested consecutive events and we refer to it several times in the discussion.

“causally” has been deleted accordingly.

A limitations paragraph is needed to offset some of the bolder claims of cause and effect. Please see the comments in the methods “controls missing” for guidance.

This comment is very much appreciated. In the revised manuscript, we provide additional behavioural data following exposure with spherical TiO₂ nanoparticles of 19.5 nm in size. The experiment was repeated three times independently and no effect on locomotion was found. The discussion now elaborates on this topic referring to publications assessing locomotor response after nanoparticle exposure.

We also dedicated a whole paragraph to elaborate on the limitations of the study (L304ff).

Methods:

Line 274-275: "A stock solution of 5 mM Mifepristone was prepared in ethanol."

This is a minor point but the grade, supplier, and strength of ethanol (100 proof; 200 proof?), should be provided.

This information is now added: 200 proof, molecular biology grade, ≥99.45%, Sigma Aldrich

Line 277: The authors have not provided an ethics/IACUC approval number or certification. The approval number should be provided so there is a record and traceback to these activities.

This information is now added:

"All work was done under approval number of the 'Netherlands Food and Consumer Product Safety Authority': 10612. As only early life stage zebrafish were used, no specific additional project authorization was needed."

Line 288-289: "The exposure concentration of 20 mg L⁻¹ PSNP was derived from an initial dose-response analysis representing a no effect concentration for malformations (data not shown)"

These data/graphs should be provided as supplementary information, not just the mortality data provided in S1. This is important information since not all malformations necessarily result in mortality and is likely a more sensitive endpoint. I don't see how the mortality graph is useful without the overlaid curve of malformations. This should be very easy to add given that the concentrations selected were reportedly chosen following analysis of this data.

This input is appreciated. However, as the exposure started at 72 hpf (after major organ development) no malformation was observed and thus not depicted in Figure S1. A reduced occurrence of swim bladder inflation was observed which is described in detail in Figure S2. The wording has been adjusted in the method section.

A concern with the data presented in S1 (and lack of detail pertaining to this in the methods) is the sample number. It appears that only a N=4 was used? Is this pooled animals per replicate as used in some later experiments? As is, this does not reflect a robust study and with no power analysis to support such a low number, I would be very concerned that the data is representative. This is particularly important since you based your future doses on these data. Having a more robust sample set here may even allow for reduced numbers later since you would have a really solid understanding of possible malformations and changes e.g., in swim bladder. The samples numbers used in every aspect of the study should be clearly reported in the methods along with the justification on the use of such low numbers. There is really no need for such a low number for the mortality and malformation dose response curves since these are lower cost experiments and fairly short in duration (2d). Please clearly state all sample numbers used in the methods long with your statistical justification.

We apologize for not having provided sufficient details which may have led to a misunderstanding that a low sample number was used. The section has been revised to provide this clarity:

"Mortality was assessed by following the protocol of the Fish Embryo Acute Toxicity Test (FET) and adapted to an exposure window from 72 to 120 hpf. Hatched larvae were exposed in 24-well plates (1 larva per well filled with 2 mL solution). Ten exposure concentrations between 10 and 100 mg L⁻¹ and a control solution consisting of egg water were tested (four replicates, ten larvae per replicate)."

The derived highest test concentration from this initial test proved to be suitable as never any mortality was observed in the follow-up assays.

Furthermore, we now state all sample numbers for every assay in the figure description and the detailed statistics in the results section following to APA style guidelines.

Controls: It is not stated whether you have an equal number of controls to experimental animals in the methods. I assume so, but I would prefer not to assume. This is captured in most sections but not all, so please go through and check that this is clear both in the text and figure legends. In the figures, the legend may say that the data is based on n=7 for example, but is this all of the data comes from 7 animals, OR each treatment has an n=7. Again I assume the latter, but this should be clarified. This would be a minor change to say “xx-animals were used in treatments and controls”.

We agree and apologize for not having provided this information at first. We have revised the manuscript and added the number of animals in the figure description and the details on the statistical analysis in the results section.

Missing controls: While an interesting study using innovative tools, I am not convinced that the effects reported in this study were a result of the plastic (polystyrene) nanoparticles, or just the presence of particles. I do not see a particle control which is really important when trying to link environmental impact and effect specifically to “plastic”. I would LOVE to see this control included because this would turn this study from good to great and would be compelling. Please consider this in future studies. I don’t doubt the “effect” observed in the present study, even though samples numbers are low in many instances. Certainly micro and nano-plastics must be doing something since it is a foreign particle that can get inside an organism (and this would cause an immune response). However, is this an effect of the nanoparticle being plastic or just being a nanoparticle that irritates the epithelium (etc.,) and is treated like a foreign particle? I don’t think you can say that “plastic” is the cause, when you haven’t controlled for other non-plastic nanoparticles. At the very least you need some discussion to justify why you did not control for the particle, then tone down your causal links in all other sections. There are no chemical analyses or plasticizer extracts used in this study either which could provide a non-particle based exposure proxy for plastic. That would be a little more challenging to be representative and likely moves into toxicity, but could have provided supportive evidence (in the absence of a particle control) of the plastic nanoparticles specifically being the cause.

We completely agree with the reviewer that a particle control study substantially improves the quality of the study presented here. Therefore, we have performed a particle control experiment using TiO₂ for the locomotion endpoint. There was no effect of TiO₂ on locomotion in the dark challenge.

The discussion is now extended on the potential adverse effect of nanoparticles indicating that our TiO₂ data is supporting a plastic particle effect but acknowledging that a particle effect cannot be entirely excluded.

Unfortunately having the varying levels of PSNPs (high, mid, low) does not help with the lack of a particle control. Perhaps you have a great justification that I have not thought of. Please include this in the discussion either as a possible limitation (if a justification is lacking), or in the methods if this was an active choice in the experimental design.

See comment above.

Insulin expression assessed by whole-mount in situ hybridization: Sample number and details needed in methods. Pooled larvae/replicate? Number of replicates?

This information is now added (L418): n = 7 per treatment group

Larval glucose measurement: Sample number and details needed in methods. Pooled larvae/replicate?

Number of replicates?

This information is now added (L425): 10 larvae per replicate (n = 5) were pooled in 55 μ l glucose buffer

Statistical analysis.

The authors should include their power analysis or related sample number determination if available.

We acknowledge this point. However, we would like to argue that there is no post-hoc power analysis needed since the significant results inherently prove that the sample size is sufficient. The locomotion assay with no significant difference is based on large sample sizes ($n > 36$) and show considerably small variance which should be arguably sufficient to support the sample size.

Other:

Genus and species names in the references should be italicized

This input is appreciated and species names have been italicized in the references.

REVIEWERS' COMMENTS:

Reviewer #1 (Remarks to the Author):

The reviewer appreciates the authors thoroughness in addressing both reviewer's concerns/comments. I do feel this work should be published. The authors need to make some edits to improve the manuscript. A more thorough discussion regarding the mechanisms of behavioural alteration is required. There is plenty of literature available to support this discussion. There is still an absence of a generalized discussion regarding neurophysiology manifesting behavioural phenotypes and mechanisms of their modulation. The authors have included only brief additions as per my previous comments. It remains unclear as to how exactly HPI-axis modulation alters behaviour in larval zebrafish (this is a very complicated subject and there is presently no clear answer). My suggestion is to discuss the relationships between the HPI-axis and aminergic systems and their signalling and how this may offer some explanation for the observed behavioural modulation (with further future studies). This should be included in addition to the suggestion that energy metabolism may be involved. Without this, it is not exactly clear what the authors conclusions are. The authors state that glucose metabolism and elevated cortisol manifest behavioural alterations, but not how this is happening.

Line 31 – 32 – “. . . suggesting that the adverse effect of PSNPs are mediated by Gr activation. . .”

I would consider re-wording this statement to allow for other adverse effects (i.e., not behavioural) that may not be mediated by Gr activation. For example, “suggesting that the behavioural effects of PSNPs are at least in-part mediated by Gr activation. . .” or something along those lines.

Line 73 – acronym for glucocorticoid receptor defined as “GR”, but written as “gr” or “Gr” elsewhere in the manuscript.

Line 236 – “increase in cortisol secretion culminate in behavioural changes”

The reviewer is still not fully convinced there is clear enough evidence to support that behavioural changes are a direct result of changes in glucose homeostasis and increase in cortisol secretion, although it may very well be the case. Maybe a simple change in word choice would satisfy the reviewer. For example, the authors could replace “culminate in” with “coincide with”.

Lines 276 – 277 – “We observed PSNP-induced alterations of cortisol levels to result in an altered behavioural response of the zebrafish larvae”

Same as previous comment. A simple solution is to re-word the sentence. My concern is in the present study it is not clear how many modes of action PSNPs have, or if a single mode of action can produce multiple effects (some responsible for HPI axis stimulation and some responsible for behavioural alteration). Including a more conservative conclusion that altered cortisol levels likely play a part in the behavioural modulation (i.e., allows for possibility of other mechanisms) seems more appropriate.

Line 291 – 294 – “It is likely that elevated cortisol increased locomotor activity either by interfering with the neural system leading to aberrant stress-coping styles (stress recovery patterns and anxiety-related behaviours)²¹ or by mobilizing glucose and thereby energy to sustain the movement¹⁸.”

I would include an additional statement indicating that further study is required to confirm this. The reviewer may be mistaken, but it appears in figure 5 that the authors tested a glucose-only batch of

fish. How do these glucose-treated fish compare to control fish? If the authors statistically compared the effect of only glucose on larval locomotor activity it could provide support, or refute, this statement.

Reviewer #2 (Remarks to the Author):

Changes made based on the initial review are acceptable and address the concerns.

REVIEWERS' COMMENTS:

Reviewer #1 (Remarks to the Author):

The reviewer appreciates the authors thoroughness in addressing both reviewer's concerns/comments. I do feel this work should be published. The authors need to make some edits to improve the manuscript. A more thorough discussion regarding the mechanisms of behavioural alteration is required. There is plenty of literature available to support this discussion. There is still an absence of a generalized discussion regarding neurophysiology manifesting behavioural phenotypes and mechanisms of their modulation. The authors have included only brief additions as per my previous comments. It remains unclear as to how exactly HPI-axis modulation alters behaviour in larval zebrafish (this is a very complicated subject and there is presently no clear answer). My suggestion is to discuss the relationships between the HPI-axis and aminergic systems and their signalling and how this may offer some explanation for the observed behavioural modulation (with further future studies). This should be included in addition to the suggestion that energy metabolism may be involved. Without this, it is not exactly clear what the authors conclusions are. The authors state that glucose metabolism and elevated cortisol manifest behavioural alterations, but not how this is happening.

We appreciate this comment and have extended the discussion on how the HPI-axis and aminergic system influences behavior (L312-322). We also invite the reader to further explore the underlying mechanisms in future studies.

In summary, the potential involvement of neurological and/or metabolic mechanisms in the observed response is indicated throughout the manuscript.

Introduction (L70-78):

“The mechanistic underpinning of these PSNP-induced behavioural changes in fish remains to a large extent elusive but can potentially be tied to neurological or metabolic effects. For example, the shoaling behaviour is thought to be mediated by neurotransmitters, specifically the dopaminergic system¹⁷. While differences in metabolic rate is a widely accepted proxy for stress response and are correlated with behavioural endpoints such as exploration or swimming activity.^{18,19} Furthermore, coping with stress displays different patterns in both serotonergic activity²⁰ and cortisol²¹ as part of a complex set of feedback interactions between the hypothalamus, the pituitary gland and interrenal tissues (HPI-axis).”

Results (L174-177):

“An increase in activity can be triggered by a variety of mechanisms including modulation of neuronal activity and rapid elevation of plasma cortisol^{19,20}. Moreover, stress-induced increases in cortisol levels can fuel modulation of neuronal activity^{19,35-37}.”

Discussion (L315-345):

“We consider two mechanisms that are most likely underlying these cortisol driven behavioral alterations. First, such hormone modulations due to PSNP exposure can interfere with the electrical activity of brain cells to alter the level of important molecules, including neurotransmitters, enzymes, or receptors. Glucocorticoid effects on the brain are highly complex and brain region-, dose- as well as time-dependent³⁵⁻³⁷. Recently such interferences with the neural system have been observed for fish like trout, medaka, and zebrafish⁴⁷⁻⁴⁹ and could potentially lead to aberrant stress-coping mechanisms (e.g. stress recovery patterns and anxiety-related behaviours)²¹. In support of this, we observed that neurotransmitter activity might indeed be affected as the gene transcribing the membrane protein that transports the neurotransmitter serotonin (slc6a4) is downregulated (Supplementary Figure 3). Second, altered glucose levels can fuel cortisol secretion with inherent changes to energy metabolism and availability to sustain activity. In addition to the contribution of Gr

activation to the observed heightened locomotor activity during the dark challenge, we explored the contribution of a dysregulated metabolic rate. The addition of 40 mM glucose as known reducer of pck1 expression in larval zebrafish⁵⁰ coincided with diminished cortisol levels (Figure 2b) and hyperactivity in larvae exposed to PSNP (Figure 4), providing strong support for distorted energy metabolism as mechanism fueling the behavioral responses in this study. While future research should uncover whether the elevated cortisol increased locomotor activity observed in this study results from interference with the neural system, increased mobilization of glucose to sustain the movement¹⁸, or a combination thereof, our results thus point towards a bottom-up driven chain of events where decreased glucose levels fuel cortisol secretion and an aberrant behavioural response (conceptually depicted in Figure 5)."

Line 31 – 32 – “. . . suggesting that the adverse effect of PSNPs are mediated by Gr activation. . .”

I would consider re-wording this statement to allow for other adverse effects (i.e., not behavioural) that may not be mediated by Gr activation. For example, “suggesting that the behavioural effects of PSNPs are at least in-part mediated by Gr activation. . .” or something along those lines.

We agree with the suggested re-wording and have made amendments accordingly.

Line 73 – acronym for glucocorticoid receptor defined as “GR”, but written as “gr” or “Gr” elsewhere in the manuscript.

We understand that this may be confusing to the reader. In general, we followed the guidelines for human gene nomenclature and its adaptation for vertebrate model organisms which is widely accepted by the research communities.

According to the guidelines, gene symbols are italicized whereas protein symbols are not italicized. Further, the capitalization of the gene or protein symbol indicates the species from which the data originates:

- *Human = all uppercase,*
- *mouse = only the first letter in upper-case,*
- *zebrafish = gene symbols have all letters in lower-case, protein symbols have the first letter in upper-case*

For clarification, we provide the following explanation in L73:

zebrafish protein: Gr, zebrafish gene: gr

Line 236 – “increase in cortisol secretion culminate in behavioural changes”

The reviewer is still not fully convinced there is clear enough evidence to support that behavioural changes are a direct result of changes in glucose homeostasis and increase in cortisol secretion, although it may very well be the case. Maybe a simple change in word choice would satisfy the reviewer. For example, the authors could replace “culminate in” with “coincide with”.

Suggestion is appreciated and amendment made accordingly.

Lines 276 – 277 – “We observed PSNP-induced alterations of cortisol levels to result in an altered behavioural response of the zebrafish larvae”

Same as previous comment. A simple solution is to re-word the sentence. My concern is in the present study it is not clear how many modes of action PSNPs have, or if a single mode of action can produce multiple effects (some responsible for HPI axis stimulation and some responsible for behavioural

alteration). Including a more conservative conclusion that altered cortisol levels likely play a part in the behavioural modulation (i.e., allows for possibility of other mechanisms) seems more appropriate.

This comment is appreciated and we agree that multiple modes of actions may be involved. We have re-worded the sentence in L276 and acknowledge the potential of multiple modes of actions in the concluding paragraph.

Line 291 – 294 – “It is likely that elevated cortisol increased locomotor activity either by interfering with the neural system leading to aberrant stress-coping styles (stress recovery patterns and anxiety-related behaviours)²¹ or by mobilizing glucose and thereby energy to sustain the movement¹⁸.”

I would include an additional statement indicating that further study is required to confirm this. The reviewer may be mistaken, but it appears in figure 5 that the authors tested a glucose-only batch of fish. How do these glucose-treated fish compare to control fish? If the authors statistically compared the effect of only glucose on larval locomotor activity it could provide support, or refute, this statement.

We appreciate this comment and are happy to indicate that the unknown mechanisms warrant further investigations.

Reviewer #2 (Remarks to the Author):

Changes made based on the initial review are acceptable and address the concerns.